# Species-Specific and Altitude-Induced Variation in Karst Plants' Use of Soil Dissolved Inorganic Carbon

Sen Rao [1,2] and Yanyou Wu [2,*]

[1] College of Life Sciences and Oceanography, Shenzhen University, Shenzhen 518060, China
[2] State Key Laboratory of Environmental Geochemistry, Institute of Geochemistry, Chinese Academy of Sciences, Guiyang 550081, China
[*] Correspondence: wuyanyou@mail.gyig.ac.cn

**Abstract:** Root-derived carbon sources supporting photosynthesis have been demonstrated to contribute to plant carbon gain in many laboratory experiments. However, it remains largely unknown whether and to what extent soil dissolved inorganic carbon (DIC) influences leaf photosynthesis in karst habitats characterized by alkaline soils with low water content. We explored this relationship by measuring the concentrations and carbon isotope signals ($\delta^{13}$C) of soil DIC, as well as the $\delta^{13}$C of water-soluble organic matter ($\delta_{WSOM}$) in leaves of nine woody species across an altitudinal gradient in karst habitats. The $\delta_{WSOM}$ varied among species by 7.23‰ and deviated from the $\delta^{13}$C of photosynthates solely assimilated from atmospheric $CO_2$ ($\delta_A$) by 0.44–5.26‰, with a mean value of 2.20‰. This systematical discrepancy ($\delta_A - \delta_{WSOM}$) could only be explained by the contribution of soil DIC to leaf total photosynthesis ($f_{DIC\_soil}$). The average values of $f_{DIC\_soil}$ considerably varied among the nine species, ranging from 2.48% to 9.99%, and were comparable with or slightly lower than those of previous laboratory experiments. Furthermore, the $f_{DIC\_soil}$ of two species significantly increased with altitude, whereas another species exhibited an opposite pattern, suggesting a highly spatial heterogeneity of DIC utilization. The present study improved our understanding of how plants adapt to the alkaline–drought soil conditions of karst habitats and thus acquire additional carbon for growth.

**Keywords:** soil dissolved inorganic carbon; photosynthesis; water-soluble organic matter; stable isotope; contribution

## 1. Introduction

Karst is a distinctive topography developed in regions with soluble bedrocks, such as limestone, dolomite, and gypsum [1], and covers about $22 \times 10^6$ km² of the earth's surface [2]. The karst habitats are characterized by thin soil layers, high content of bicarbonate and high pH in soils, and less vegetation cover [3–5]. For plants growing in karst habitats, water demand for maintaining the basic functions is often restricted due to the shallow soil [6]. For example, drought stress can lead to the decline of stomata opening and photosynthesis and downregulated metabolic processes [7]. A prolonged drought will result in hydraulic failure and/or carbon starvation, increasing the risk of mortality [8,9]. On the other hand, the alkaline soil condition in karst habitats is unfavorable for plant uptake of nutrients, such as phosphate, ferrous iron, and zinc [10,11]. As a consequence, the vegetation productivity of the karst ecosystem is usually much less than that in nonkarst regions [12]. Although plants can adopt tolerance and/or avoidance strategies to cope with environmental stress [13,14], it does not fully explain why some species grow well in karst habitats. Hence, other physiological processes of karst plants are suspected to play an important role in regulating their growth under dry and alkaline soil conditions.

Recently, root-derived carbon sources supporting photosynthesis have reemerged as a hotspot in plant ecophysiological studies [15–18]. Research shows that $^{13}$C-enriched $CO_2$

or $HCO_3^-$ labeling in the root zones can be transported upward through the transpiration stream [19] and thus affects the carbon isotope composition ($\delta^{13}C$) of leaf photoassimilates and aboveground tissues [20,21]. Although these results are obtained from the manipulated experiments inside a laboratory, it may be applied to karst plants grown in the field, given the high concentration of bicarbonate contained in karst soils. The additional carbon gain may also compensate for the decrease in the fixation of atmospheric $CO_2$ induced by drought stress [22,23]. However, to the best of our knowledge, no research has been reported to investigate plants' use of dissolved inorganic carbon (DIC, mainly including dissolved $CO_2$ and $HCO_3^-$) in karst habitats. Although many hydroponic experiments were designed to simulate karst soil conditions, for example, bicarbonate stress (usually more than 10 mM) [24,25], osmotic stress (simulating water scarcity) [26], and their interactions [26], their results do not necessarily reflect the utilization of soil DIC by plants due to the differences in the microenvironment of the root zones between soil conditions and hydroponic solutions [27]. Hence, there is a need to fill this knowledge gap between laboratory experiments and field trials.

Appropriate methods and models for quantification are critical to understanding plants' use of soil DIC. Currently, two methods are available to estimate this utilization. Method (i) applies the high abundance $^{13}C$ labeling and then calculates the ratio of soil DIC fixed in specific tissues or organs to the total carbon gain. The total carbon gain includes soil DIC fixation, apparent photosynthesis measured by commercial infrared gas analyzers, and respiration [20,28]. However, this calculation is not directly linked to photosynthesis. Method (ii) uses a two-source $^{13}C$ labeling in combination with isotope mixing models to determine the contribution of root-derived DIC to leaf total photosynthesis [22–24,29]. Nevertheless, uncertainties remain associated with the photosynthetic $^{13}C$ discrimination and post-photosynthesis processes. Additionally, both labeling methods depend on irrigation or hydroponic culture, which will inevitably change the soil conditions, especially in karst environments. The labeling experiment also requires a sealed environment and has a high cost, which does not apply to large-scale field experiments. For these reasons, a new approach needs to be developed to calculate the contribution of soil DIC to leaf total photosynthesis (see Section 2). It has been shown that the natural abundance of $^{13}C$ has the potential to address a wide range of ecophysiological and biochemical questions [30]. For example, naturally occurring $\delta^{13}C$ signals in plant tissues integrate plant–environment interactions over long periods [31]. In $C_3$ plants, $\delta^{13}C$ of leaves is mainly controlled by photosynthetic $^{13}C$ discrimination [32], which can be altered by stomatal control and the activity of the carboxylation enzyme, Rubisco. Furthermore, when other carbon sources are supplying the leaf photosynthesis, for instance, uptake of DIC from the xylem sap of the host by the mistletoe or utilization of root-derived DIC by some plants [26,33], the $\delta^{13}C$ of leaves can also be modified.

In karst habitats, altitude plays an important role in affecting the soil conditions and microclimate [34], which determine the distribution of plant species [35]. Species with different life forms may vary in functional traits, such as root uptake and xylem transport of water and DIC, thus influencing the extent of soil DIC used by plants. In addition, the concentration of soil DIC may change with altitudes, which probably acts on the proportion of soil DIC used by leaf photosynthesis. In this study, the primary objective was to investigate whether and to what extent soil DIC influences the leaf photosynthesis of karst plants at different altitudes. Based on our previous observation [26], we expected to find a large discrepancy between the measured $\delta^{13}C$ of photosynthates and that predicted by a photosynthetic $^{13}C$ discrimination model, which indicates the involvement of soil DIC in leaf photosynthesis. We hypothesized that the contribution of soil DIC to leaf total photosynthesis was species-specific due to the different physiological responses (e.g., leaf gas exchange) to the environmental factors. Our third hypothesis is the altitude exerts an influence on plants' use of soil DIC, given the differences in soil conditions and microclimate induced by altitude.

## 2. Materials and Methods

### 2.1. Study Site

The field experiments were conducted in August 2016 on Mt. Sanmao, Nanming District, Guiyang, southwest China (26°33′57″ N, 106°44′56″ E), with an elevation between 1080 and 1315 m above sea level. In this region, the annual mean precipitation and mean air temperature during 2011 and 2015 were 1129.5 mm and 15.3 °C, respectively. The mean minimum temperature of the coolest month (January) was 4.6 °C, whereas the mean maximum temperature of the warmest month (July) ranged from 25 to 28 °C. Rainfall in August was decreased compared with that in July. The study site was a typical karst landform covered with a secondary forest. Soils under the vegetation examined were alkaline (pH 7.09–7.75), rich in organic matter and bicarbonate, and have low soil water content according to our preliminary survey at the same site [34]. Moreover, these soil properties changed along an altitude gradient. The bedrock was mainly carbonatite, which developed shallow calcareous soil. The thickness of the soil layer varied between 27 and 58 cm. The vegetation community was mixed with evergreen and deciduous broad-leaved forest, with the canopy height in the range of 0.8–7.6 m.

### 2.2. Experimental Design

Six plots were established along three altitudes: S1 and S2 on the lower altitude, S3 and S4 on the medium altitude, and S5 and S6 on the higher altitude (Figure 1). Each plot had an area of $20 \times 20$ m$^2$ and 60–80 m away from the neighboring plot at each altitude. Common species in each plot were recorded in Table 1. Collectively, four tree species (*Ligustrum lucidum* ait., *Broussonetia papyrifera* L., *Platycarya longipes* Wu., and *Zelkova serrata* (Thunb.) Makino) and five shrub species (*Viburnum dilatatum* Thunb., *Ampelopsis delavayana* Planch., *Rosa cymosa* Tratt., *Zanthoxylum armatum* DC., and *Rubus biflorus* Buch.-Ham. ex Smith) were selected for the subsequent measurement and sampling. Species in each plot included four replicates (individuals).

**Table 1.** Common species in plots of different altitudes. S1–S6 represented six plots established at three altitudes.

| Altitudes | Sampling Sites | Elevation (m) | Species | |
|---|---|---|---|---|
| | | | Tree | Shrub |
| Lower | S1 | 1115 | *L. lucidum, B. papyrifera* | *V. dilatatum, A. delavayana, R. cymosa* |
| | S2 | 1123 | *L. lucidum, B. papyrifera* | *A. delavayana, R. cymosa* |
| Medium | S3 | 1224 | *P. longipes, Z. serrata* | *V. dilatatum, A. delavayana* |
| | S4 | 1229 | *L. lucidum, P. longipes, Z. serrata* | *A. delavayana, R. cymosa* |
| Higher | S5 | 1289 | *P. longipes* | *V. dilatatum, A. delavayana, Z. armatum, R. biflorus* |
| | S6 | 1292 | *L. lucidum* | *A. delavayana, Z. armatum, R. biflorus* |

### 2.3. Leaf Gas-Exchange Measurements

Leaf gas exchange was measured in the morning (9:00–11:30 a.m.) of three sunny days (17–19 August). The net photosynthetic rate (*A*), stomatal conductance ($g_s$), transpiration rate (*E*), ratio of intercellular to the ambient partial pressure of $CO_2$ ($c_i/c_a$), and instantaneous water use efficiency ($WUE_i$) of the fully expanded leaves were measured using an infrared gas analyzer LI-6400 (Li-Cor, Lincoln, NE, USA), equipped with a clear topped $2 \times 3$ cm cuvette. The block temperature of the leaf chamber was set to be close to the air temperature. Near-ambient air at the height of 5 m above the ground was supplied to the leaf chamber at a rate of 500 µmol s$^{-1}$ in the sample cell of LI-6400. Leaves were placed inside the cuvette for 2–3 min until steady state, and then the gas exchange parameters were manually logged. In addition, dark respiration of leaves was measured after 30 min of dark adaptation. Three leaves were measured per tree or shrub. At the end of the measurements, leaf samples were immediately collected and stored in a cooler box for further determination.

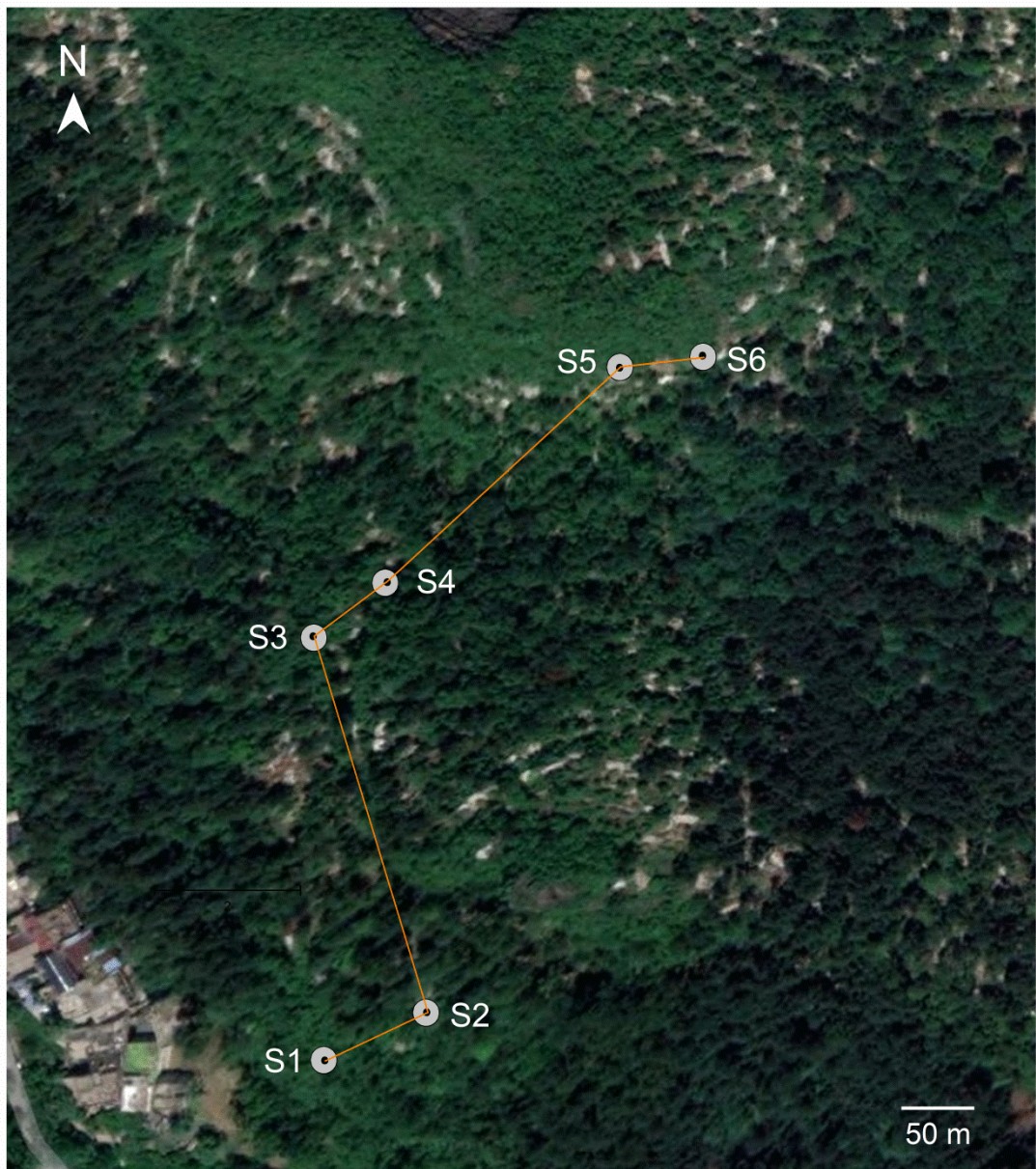

**Figure 1.** Location of sampling sites (S, circles) near Nanming District, Guiyang, southwest China (adapted from Google Map).

### 2.4. Air and Soil Sampling

Ambient air at the height of 5 m above the ground was sampled once in each plot during the measurement of leaf gas exchange. The air sample was pumped into gas sampling bags (Dalian Delin Gas Packing Co., Ltd., Dalian, China) for further analysis. Soil air was sampled in the root zone of each plant. Three days prior to our sampling, two holes of 10–15 cm depth with 2 cm diameter were drilled into the soil. Subsequently, two soft gas-piping tubes with filters were inserted into the holes, and then soil air was sucked out. Finally, the end of the tubes was clamped with clips. These processes allowed a long time for soil air to equilibrate between soil porosity and the internal space of tubes. At the day of sampling, the litter layer was removed, and then soil air was pumped into a gas sampling bag with a volume of 30 mL through one of the tubes, waiting for further determination. Another tube was connected to LI-6400, and, thus, the $CO_2$ concentration of soil air was directly measured when it reached a peak value. Soil samples (approximately

150 g) were collected from the top layer of soil (0 to 20 cm) after removing the litter. They were immediately kept in sealed plastic bags and stored in a cooler box for further analysis.

### 2.5. Determination of Carbon Isotope Composition

The $\delta^{13}C$ of water-soluble organic matter (WSOM) in leaves instead of bulk leaves was determined to represent the carbon isotope signals of photosynthates. Leaf WSOM has a high turnover rate [36–38] and mainly includes soluble carbohydrates, the hydrolysates of starch, and amino acids [39,40]. The extraction of WSOM referred to the protocols of [41]. Firstly, dried leaf material was ground, and 50 mg sample was put into a 2.5 mL centrifuge tube. After 1 mL of double-deionized water was added, the tube was agitated and then placed in a freezer for 1 h at 4 °C. Secondly, the samples were heated for 10 min at 95 °C, cooled to room temperature (RT), and centrifuged for 10 min at 12,000 g. Thirdly, an aliquot of 25 μL of the supernatant was transferred to a tin capsule and dried at 60 °C for 24 h. The $\delta^{13}C$ of the WSOM was analyzed with an Isotope Ratio Mass Spectrometer (MAT-253, Thermo Fisher Scientific Inc., Waltham, MA, USA). Three leaf samples were measured per plant. The carbon isotopic ratio of the samples was calculated as $\delta^{13}C$ (‰) = [($R_{sample}$ /$R_{standard}$) − 1] × 1000, where $R_{sample}$ and $R_{standard}$ are the $^{13}C/^{12}C$ ratio of the sample and Vienna Pee Dee Belemnite (VPDB), respectively. The standard deviation of repeated measurements for MAT253 was 0.1‰ for $\delta^{13}C$.

The $\delta^{13}C$ of atmospheric and soil $CO_2$ was determined by TG-IsoPrime (GV. Instruments Ltd., Manchester, UK). The method for soil DIC extraction was slightly modified from that of [42]. Briefly, 100 g of soil was placed into a 500 mL glass bottle and then mixed with 200 mL of $CO_2$-free water. The mixture was homogeneously stirred for 5 min and then incubated at RT for 30 min. Subsequently, the supernatant was centrifuged for 15 min at 3500× $g$ and then filtered through a 0.45 μm water-base membrane. The total alkalinity of the filtrate was detected by an Aquamerck alkalinity kit (MColortest, Merck KGaA, Darmstadt, Germany). At the same time, soil pH was measured using a pH meter (S210-B, Mettler Toledo Group Ltd., Zurich, Switzerland). The remaining filtrate was transferred to a 50 mL bottle and sealed with a rubber plug. The air inside the bottle was drawn by a vacuum pump. Subsequently, the filtrate of DIC was pretreated with an aliquot of 3 mL phosphoric acid to react with DIC and produce $CO_2$. The generated $CO_2$ was then processed with a custom-built vacuum extraction system, through which the $CO_2$ was purified. The $\delta^{13}C$ of the $CO_2$ generated from soil DIC was then analyzed by MAT-252 (Finnigan, Bremen, Germany). The measurement precision for TG-IsoPrime and MAT252 were 0.1‰ and 0.01‰ for $\delta^{13}C$, respectively.

### 2.6. Quantification of the Utilization of Soil DIC by Plants

We accounted for two end members, namely atmospheric $CO_2$ and soil DIC, that might contribute to the constitution of newly formed photosynthates as well as its $\delta^{13}C$ [26]. The $\delta^{13}C$ of photosynthates ($\delta_A$) solely assimilated from atmospheric $CO_2$ could be predicted with Equation (1) [43]:

$$\delta_A = \frac{\delta_a - \Delta^{13}C_{com}}{1 + \Delta^{13}C_{com}/1000} \tag{1}$$

where $\delta_a$ is the carbon isotope composition of atmospheric $CO_2$, whereas $\Delta^{13}C_{com}$ is the comprehensive discrimination against $^{13}C$ including the diffusion of $CO_2$ across the boundary layer, intercellular air space, mesophyllcell, as well as the contributions of respiration and photorespiration. $\Delta^{13}C_{com}$ is calculated using Equation (2) [44]:

$$\Delta^{13}C_{com} = \frac{1}{1-t}\left[\bar{a}\frac{c_a - c_i}{c_a}\right] + \frac{1+t}{1-t}[a_m\frac{c_i - c_c}{c_a} + \frac{c_c}{c_a}(b_3{}' - \frac{\alpha_b}{\alpha_e}e\frac{R_d}{V_c} - \frac{\alpha_b}{\alpha_f}f\frac{F}{V_c})] \tag{2}$$

where $c_a$, $c_i$, and $c_c$ denote the $CO_2$ partial pressure in the ambient air of the cuvette, intercellular air, and chloroplast, respectively; $a_m$ is the summed $^{12}C/^{13}C$ fractionation during the dissolution of $CO_2$ and liquid-phase diffusion ($a_m$ = 1.8‰); $b_3{}'$ is the $^{12}C/^{13}C$

fractionation of Rubisco ($b_3' = 29‰$, [45]); $e$ is the $^{12}C/^{13}C$ fractionation for day respiration ($e = 0$, [46]); and $f$ is the $^{12}C/^{13}C$ fractionation during photorespiration ($f = 11‰$, [47]); $R_d$, $V_c$, and $F$ are nonphotorespiratory $CO_2$ released in the dark, Rubisco carboxylation rate for the fixed atmospheric $CO_2$, and photorespiratory rate, respectively; $\alpha_b = 1 + b_3'$, $\alpha_e = 1 + e$, and $\alpha_f = 1 + f$; $t$ and $\bar{a}$ are the ternary correction factor and the weighed fractionation for $CO_2$ diffusion across the boundary layer and stomata, respectively, which are calculated as [32]

$$t = \frac{\alpha_{ac} E}{2 g_{ac}} \tag{3}$$

$$\bar{a} = \frac{a_b(c_a - c_s) + a_s(c_s - c_i)}{c_a - c_i} \tag{4}$$

where $\alpha_{ac} = 1 + \bar{a}$, $E$ is the transpiration rate; $g_{ac}$ is the combined boundary layer and stomatal conductance to $CO_2$; $a_b$ and $a_s$ are the $^{12}C/^{13}C$ fractionation for $CO_2$ diffusion across the boundary layer and through the stomata ($a_b = 2.9‰$, $a_s = 4.4‰$), respectively; and $c_s$ is the $CO_2$ partial pressure at the leaf surface. $c_c$ was not directly measured by LI-6400, but it was calculated with Equation (5) [37]:

$$A = g_i(c_i - c_c) \tag{5}$$

where $g_i$ is an internal $CO_2$ conductance. In this study, we assumed $g_i$ of 0.5 mol m$^{-2}$ s$^{-1}$ for woody species according to [48]. $V_c$ and $F$ are calculated as [44]

$$V_c = A + R_d + F \tag{6}$$

where

$$F = \frac{\Gamma^*(A + R_d)}{c_c - \Gamma^*} \tag{7}$$

where $\Gamma^*$ is the $CO_2$ compensation point in the absence of $R_d$, which was fitted using Equation (8) [49]:

$$\Gamma^* = e^{\left(c - \frac{\Delta H * 1000}{(R * (273 + T_{leaf}))}\right)} \tag{8}$$

where $c$, $\Delta H$, and $R$ are the scaling constant (13.49), energy of activation (24.46 kJ K$^{-1}$ mol$^{-1}$), and molar gas constant (0.008314 kJ J$^{-1}$ mol$^{-1}$), respectively.

When accounting for the involvement of soil DIC in leaf photosynthesis, the carbon isotopic composition of photosynthate ($\delta_A'$) originating from the assimilation of soil DIC is given by

$$\delta_A' = \frac{\delta_{DIC} - \Delta^{13}C_{DIC}}{1 + \Delta^{13}C_{DIC}/1000} \tag{9}$$

where $\delta_{DIC}$ is the carbon isotope composition of soil DIC collected from root zones of each plant; $\Delta^{13}C_{DIC}$ is the photosynthetic discrimination against $^{13}C$ combining the carboxylation of DIC and the effects of respiration and photorespiration, which is expressed as

$$\Delta^{13}C_{DIC} = b_3' - \frac{\alpha_b}{\alpha_e} e \frac{R_d}{V_c'} - \frac{\alpha_b}{\alpha_f} f \frac{F}{V_c'} \tag{10}$$

where $V_c'$ is the rate of Rubisco carboxylation for DIC and given by

$$V_c' = A_{DIC\_soil} + R_d + F \tag{11}$$

where $A_{DIC\_soil}$ is the net photosynthetic rate for soil DIC. Empirically, $A_{DIC}$ is less than 10% of $A$ in many species. Here, we assumed an initial value for $A_{DIC\_soil}$ ($A_{DIC\_soil0}$) to be $A/10$. Although this assumption introduced some errors, it had little effect on the calculation of $\Delta^{13}C_{DIC}$ and the contribution of DIC to leaf total photosynthesis. Considering bicarbonate was the dominant species in soil DIC under high pH (7.09–7.75), $\delta_{DIC}$ in Equation (9) is

derived from the $\delta^{13}C$ of bicarbonate minus the discrimination related to the conversion between bicarbonate and $CO_2$ ($e_b'$, 9‰ at 25 °C).

However, studies showed that root-respired $CO_2$ ($C_{\_R}$) can also dissolve in the xylem sap [50]. $C_{\_R}$ is primarily reported to range from 0.01 to 10.98 mM and varies with species, height, season, etc. [51,52]. Therefore, we proposed a ratio of the concentration of soil $CO_2$ or DIC ($C_{\_DIC}$) to $C_{\_R}$ in the xylem, $f_{s/x} = C_{\_DIC}/(C_{\_DIC} + C_{\_R})$. In the present study, we directly measured $C_{\_DIC}$ and adopted the mean value of 5 mM for $C_{\_R}$. Thus, the $\delta^{13}C$ of the mixed DIC in the xylem sap ($\delta_{DIC\_xylem}$) is calculated as

$$\delta_{DIC\_xylem} = f_{s/x} \cdot \delta_{DIC} + (1 - f_{s/x})\, \delta_R \tag{12}$$

where $\delta_R$ is the carbon isotope composition of root-respired $CO_2$ dissolved in the xylem sap. We assumed that the respiration of living tissues came from the recently fixed C, such as WSOM, which would carry the enriched isotopic signal to roots due to post-photosynthetic fractionation (−2‰, [53]) and then result in −1−−3‰ of respiratory fractionation in roots [54–56]. Here, we used the intermediate value −2‰ for respiratory fractionation; thus, $\delta_R$ is calculated as $\delta_R = \delta_{WSOM} + 4‰ - a_m$.

Considering that both atmospheric $CO_2$ and soil DIC could be used for leaf photosynthesis, the mixture of photosynthates from two substrates determined the value of $\delta_{WSOM}$, which is expressed as a two-end-member mixing model:

$$\delta_A' \cdot f_{DIC\_xylem} + \delta_A \left(1 - f_{DIC\_xylem}\right) = \delta_{WSOM} \tag{13}$$

where $f_{DIC\_xylem}$ is the proportion of xylem DIC contributing to $\delta_{WSOM}$; $\delta_A'$ is recalculated with Equation (9) but in which $\delta_{DIC}$ is replaced by $\delta_{DIC\_xylem}$. Finally, the contribution of soil DIC to the total leaf photosynthesis ($f_{DIC\_soil}$) is calculated as

$$f_{DIC\_soil} = f_{s/x} \cdot f_{DIC\_xylem} \tag{14}$$

### 2.7. Statistical Analysis

The comparison of mean values between species and altitudes was determined by one-way ANOVA with a *t*-test. Linear regression was used to investigate the relationship between $f_{DIC\_soil}$ and $\delta_A$-$\delta_{WSOM}$. Pearson's correlation was conducted to evaluate the relationships between physiological or environmental factors and $f_{DIC\_soil}$. All statistical analyses were performed using SPSS statistical software for Windows 21 (SPSS Inc., Chicago, IL, USA). The tests were considered significant at the $p < 0.05$ level. All data were expressed as the mean ± 1SE.

## 3. Results
### 3.1. Leaf Gas Exchange

All gas-exchange parameters exhibited significant differences within nine species (Table 2). However, no clear trend was observed between two life forms (tree and shrub). The maximum and minimum values of $A$ were 18.24 ± 0.92 μmol m$^{-2}$ s$^{-1}$ in *B. papyrifera* and 9.81 ± 0.92 μmol m$^{-2}$ s$^{-1}$ in *R. cymosa*, respectively. The $g_s$ of nine species ranged from 0.14 to 0.28 mol m$^{-2}$ s$^{-1}$, suggesting mild-to-moderate drought stress. In contrast, there were small variations of $E$, $c_i/c_a$, and $WUE_i$ among nine species, with mean values of 3.91 mol $H_2O$ m$^{-2}$ s$^{-1}$, 0.67, and 3.27 μmol $H_2O$ mol$^{-1}$, respectively.

**Table 2.** Leaf gas exchange of nine species across six plots. *A*, net photosynthetic rate; $g_s$, stomatal conductance; *E*, transpiration rate; $c_i/c_a$, ratio of intercellular to ambient partial pressure of $CO_2$; $WUE_i$, instantaneous water use efficiency. Values represent mean $\pm$ SE. N = 4 for each species in the specific plot. Capital letters indicate significant differences ($p < 0.05$) among species.

| Species | $A$ ($\mu mol\ m^{-2}\ s^{-1}$) | $g_s$ ($mol\ m^{-2}\ s^{-1}$) | $E$ ($mol\ H_2O\ m^{-2}\ s^{-1}$) | $c_i/c_a$ | $WUE_i$ ($\mu mol\ H_2O\ mol^{-1}$) |
|---|---|---|---|---|---|
| *L. lucidum* | 11.07 (0.72) C | 0.14 (0.01) D | 3.38 (0.20) CD | 0.60 (0.01) C | 3.30 (0.14) BCD |
| *B. papyrifera* | 18.24 (0.92) A | 0.28 (0.01) A | 3.67 (0.19) BCD | 0.67 (0.01) AB | 4.98 (0.13) A |
| *P. longipes* | 9.99 (0.59) C | 0.19 (0.02) BCD | 3.81 (0.16) ABCD | 0.72 (0.02) A | 2.71 (0.23) D |
| *Z. serrata* | 13.91 (0.27) B | 0.21 (0.01) BC | 3.95 (0.25) ABCD | 0.67 (0.01) AB | 3.61 (0.22) B |
| *V. dilatatum* | 10.58 (0.47) C | 0.18 (0.01) CD | 3.59 (0.11) CD | 0.70 (0.01) AB | 2.96 (0.13) CD |
| *A. delavayana* | 13.39 (0.47) B | 0.24 (0.01) AB | 4.58 (0.26) A | 0.70 (0.01) AB | 3.11 (0.18) BCD |
| *R. cymosa* | 9.81 (0.50) C | 0.16 (0.02) CD | 3.23 (0.22) D | 0.65 (0.03) BC | 3.10 (0.11) BCD |
| *Z. armatum* | 14.07 (1.03) B | 0.21 (0.02) BC | 4.18 (0.27) ABC | 0.66 (0.02) AB | 3.42 (0.24) BC |
| *R. biflorus* | 13.54 (0.41) B | 0.21 (0.01) BC | 4.46 (0.13) AB | 0.68 (0.01) AB | 3.04 (0.10) BCD |

### 3.2. Characteristics of Soil DIC and $CO_2$

The average concentrations of DIC (or $C_{\_DIC}$) in the rhizosphere of four tree species were similar (around 9.26 mM), whereas in five shrub species, they exhibited large variations, ranging from 8.10 mM in *R. cymosa* to 10.50 mM in *Z. armatum* (Figure 2A). In contrast, the mean concentrations of soil $CO_2$ varied from 3919.91 ppm in *P. longipes* to 5279.83 ppm in *B. papyrifera*, higher than the mean value of 3454.42 ppm in all shrub species. According to Henry's law, the partial pressure of $CO_2$ over a solution was proportional to the concentration of the $CO_2$ in the solution. The calculated quantity of $CO_2$ dissolved in soil water ($[CO_2^*]$) was in the range of 0.55 to 0.92 mM at pH 7 and 25 °C across nine species, much lower than the mean value of $[CO_2^*]$ in xylem sap (or $C_{\_R}$) investigated in many species. Thus, it was unlikely that soil $CO_2$ could be fixed by plant roots due to a $CO_2$ gradient from roots to the soil. In addition, the interspecies differences in the $\delta^{13}C$ of soil DIC and $CO_2$ were both less than 2‰ (Figure 2B). The mean values of soil DIC and $CO_2$ among nine species were −10.19‰ and −20.01%, respectively.

### 3.3. $\delta^{13}C$ of Photosynthates

The $\delta^{13}C$ of the newly formed photosynthates in leaves could be determined by measuring the isotope signals of WSOM, or predicted with Eqn. 1 that only accounted for the assimilation of atmospheric $CO_2$ (Figure 3A). $\delta_{WSOM}$ varied all among species by about 3–4‰ and was generally lower than $\delta_A$ in all species. The discrepancy between $\delta_A$ and $\delta_{WSOM}$ ($\delta_A$-$\delta_{WSOM}$) was species-specific. Figure 3B displayed the distribution of $\delta_A$-$\delta_{WSOM}$ in leaves of nine species under varying altitudes. The highest frequency (approximate 30%) of $\delta_A$-$\delta_{WSOM}$ was located in the range of 1 to 2‰. Furthermore, half of the $\delta_A$-$\delta_{WSOM}$ was distributed between 2‰ and 6‰. This phenomenon clearly illustrated the fact that there were other carbon sources, for example, soil DIC, which participated in leaf photosynthesis. The photosynthates originating from the mixture of soil DIC and xylem $[CO_2^*]$ carried more $^{13}C$-depleted signal ($\delta_A'$, see Supplementary Data), thus resulting in $\delta_{WSOM}$ systematically lower than $\delta_A$.

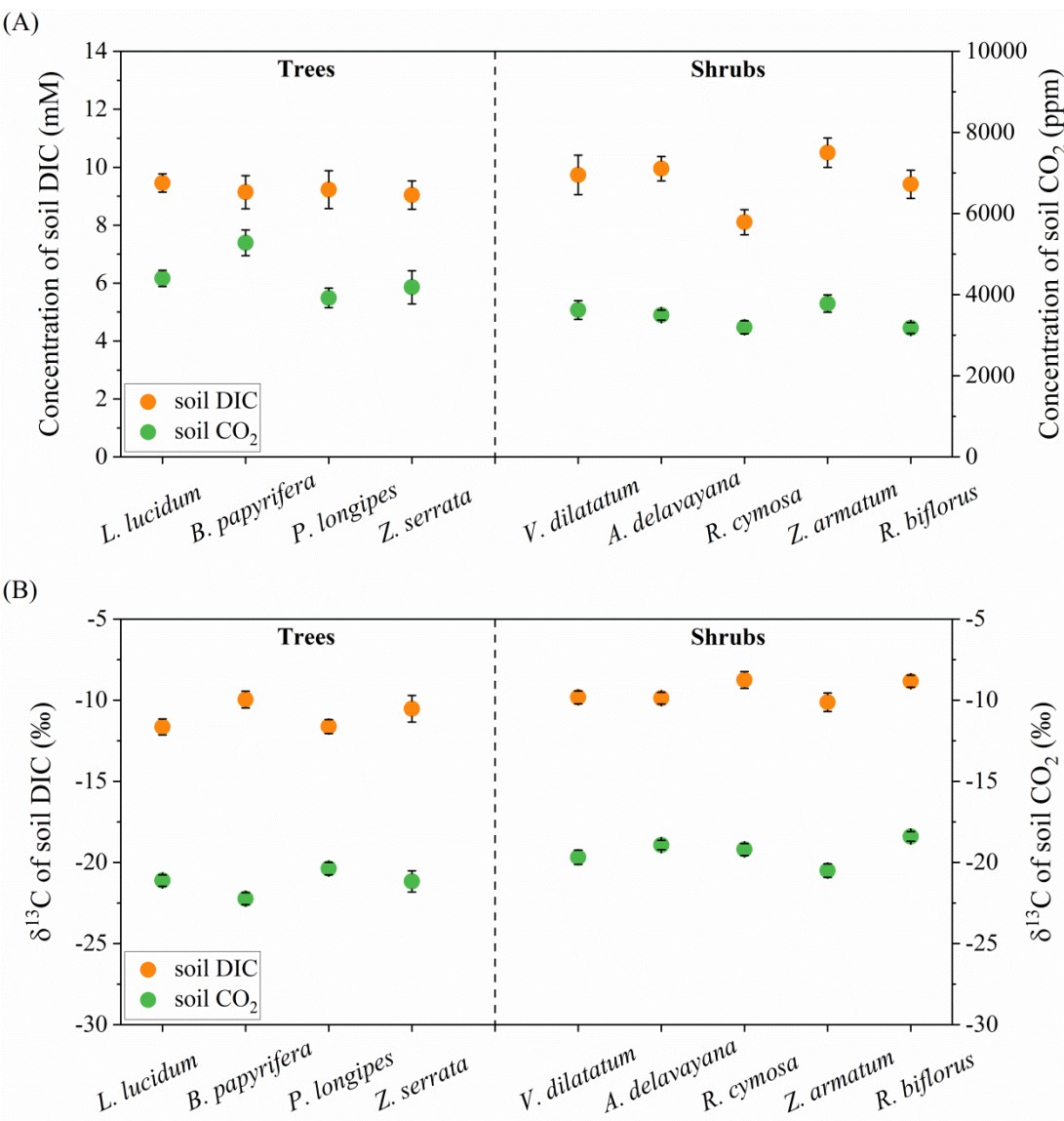

**Figure 2.** Concentrations (**A**) and $\delta^{13}$C (**B**) of soil DIC and $CO_2$ in the root zones of nine species across three altitudes. N = 4 for each species in the specific plot.

*3.4. Interspecies Difference in $f_{DIC\_soil}$*

The potential contribution of soil DIC to leaf total photosynthesis ($f_{DIC\_soil}$) considerably varied with species across the six plots (Figure 4). In tree species, the average values of $f_{DIC\_soil}$ in *L. lucidum* and *B. papyrifera* were 8.93% and 9.54%, respectively, significantly higher than those in *P. longipes* and *Z. serrata* (both less than 2.2%). By contrast, the mean values of $f_{DIC\_soi}$ within the shrub species varied from 2.48% to 9.99%, with the highest value in *Z. armatum* and the lowest value in *R. biflorus*. Furthermore, $f_{DIC\_soil}$ strongly correlated with $\delta_A$-$\delta_{WSOM}$ ($p < 0.001$), with the coefficient of determination ($R^2$) as 0.99. For the nine species, a higher value of $\delta_A$-$\delta_{WSOM}$ generally indicated a higher value of $f_{DIC\_soil}$ (Figure 5). This tight relation further demonstrated the plants' use of soil DIC in the karst habitats. Additionally, $f_{DIC\_soi}$ significantly correlated with $A$, $c_i/c_a$, and $WUE_i$ (Table 3).

(A)

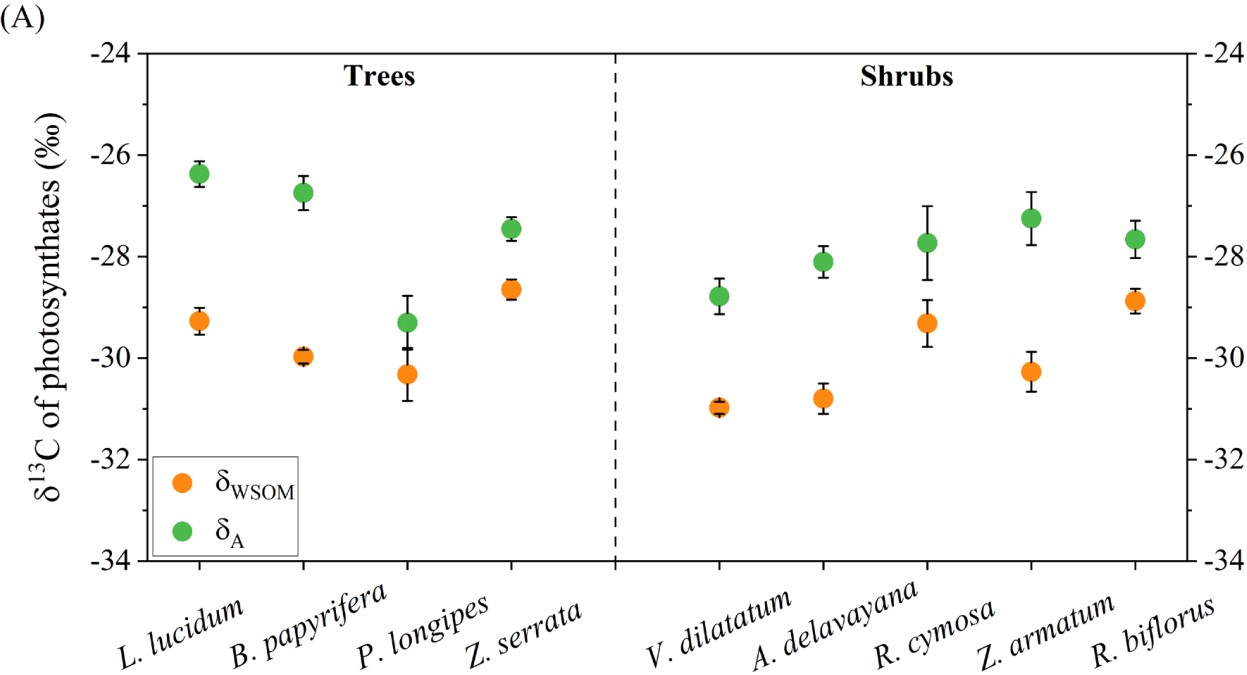

(B)

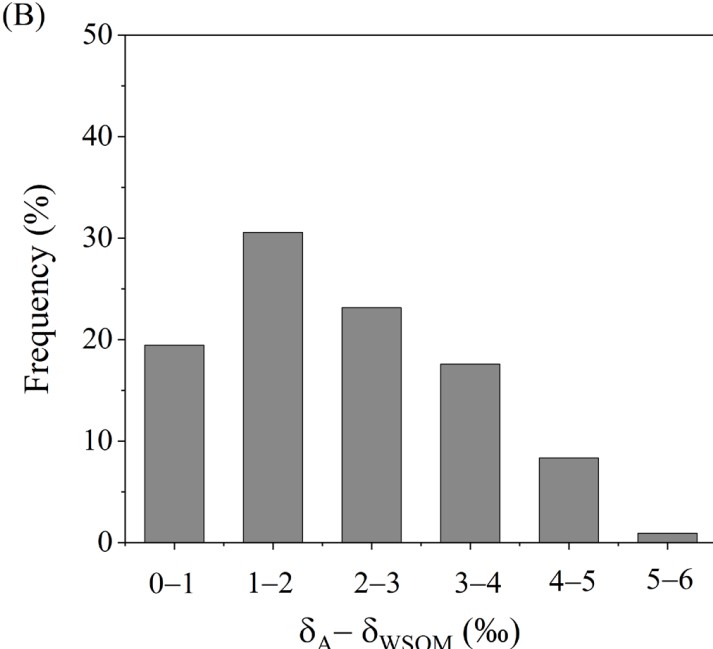

**Figure 3.** $\delta^{13}C$ of photosynthates determined by measuring isotope signals of leaf WSOM (orange closed circle) or predicted with Equation (1) that only accounted for assimilation of atmospheric $CO_2$ (green closed circle) of nine species across three altitudes (**A**); frequency distribution of the discrepancy between $\delta_A$ and $\delta_{WSOM}$ (**B**).

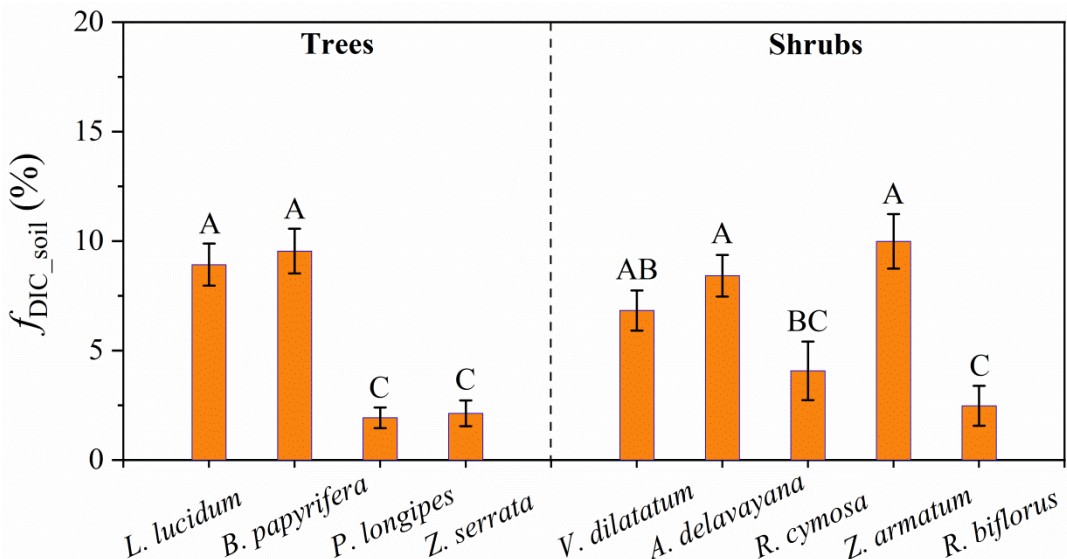

**Figure 4.** Contribution of soil DIC to leaf total photosynthesis ($f_{DIC\_soil}$) in nine species across three altitudes. N = 4 for each species in the specific plot. Capital letters on the error bar indicate significant differences ($p < 0.05$) among species.

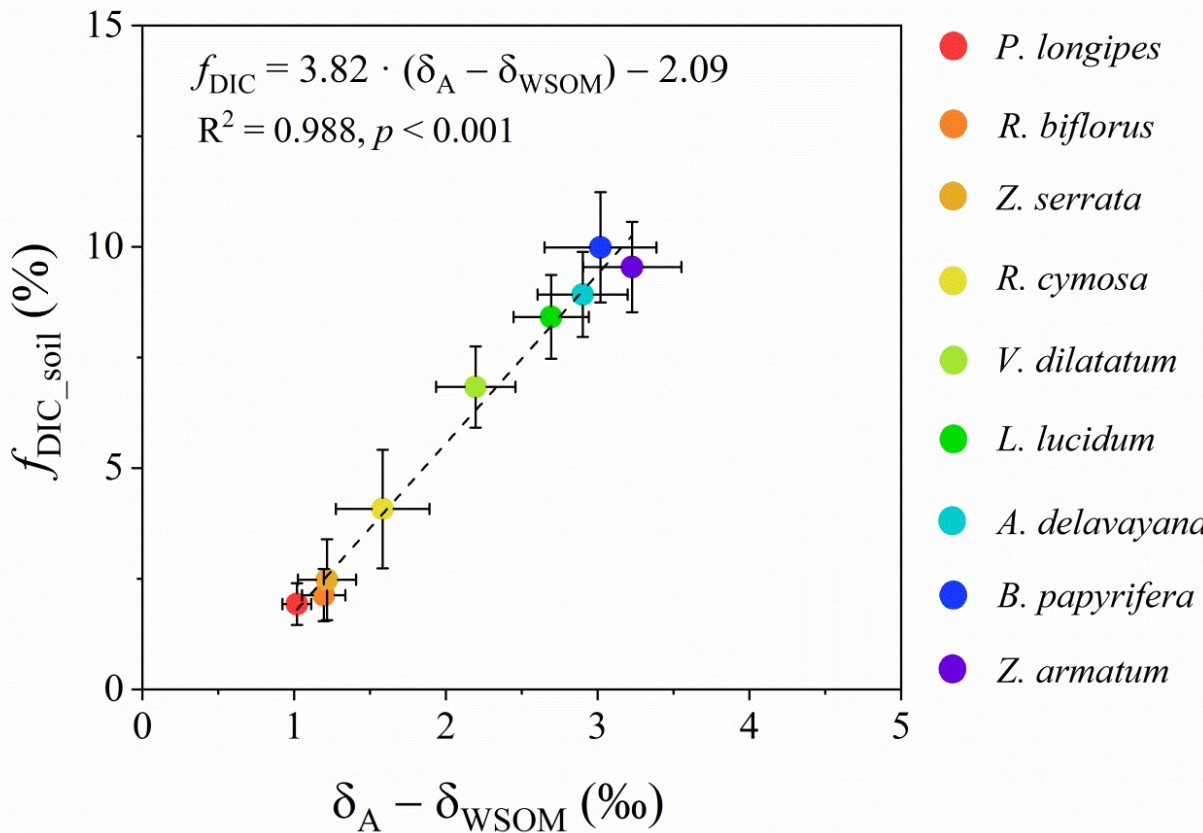

**Figure 5.** Relationship between ($\delta_A$-$\delta_{WSOM}$) and $f_{DIC\_soil}$ among nine species across three altitudes. N = 4 for each species in the specific plot.

**Table 3.** Relationships between physiological or environmental factors and $f_{\text{DIC\_soil}}$ analyzed by Pearson's correlation. N = 108.

| Factors | $f_{\text{DIC\_soil}}$ | |
|---|---|---|
| | **r** | **p** |
| $A$ | 0.287 | 0.003 |
| $g_s$ | −0.122 | 0.210 |
| $E$ | −0.096 | 0.325 |
| $c_i/c_a$ | −0.486 | <0.001 |
| $WUE_i$ | 0.350 | <0.001 |
| $C_{\text{DIC}}$ | 0.053 | 0.586 |
| $\delta_{\text{DIC}}$ | −0.100 | 0.306 |
| $\delta_{\text{WSOM}}$ | −0.155 | 0.105 |

### 3.5. Variations in $f_{\text{DIC\_soil}}$ at Different Altitudes

The impact of altitude on $f_{\text{DIC\_soil}}$ was investigated in one tree (*L. lucidum*) and two shrub species (*V. dilatatum* and *A. delavayana*), which all appeared at three altitudes (Table 1; Figure 6). In *L. lucidum* and *A. delavayana*, $f_{\text{DIC\_soil}}$ tended to increase from lower to higher altitude ($p < 0.05$), whereas in *V. dilatatum*, the pattern was opposite.

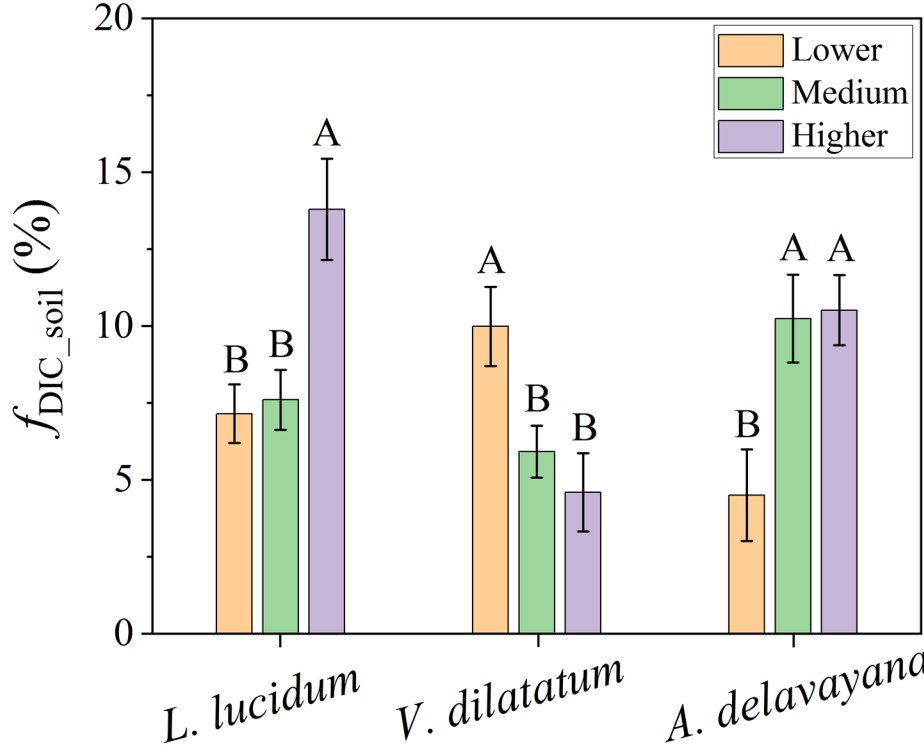

**Figure 6.** $f_{\text{DIC\_soil}}$ of one tree species (*L. lucidum*) and two shrub species (*V. dilatatum* and *A. delavayana*) at different altitudes. N = 4 for each species in the specific plot. Capital letters on the error bar indicate significant differences ($p < 0.05$) among altitudes.

## 4. Discussion

### 4.1. Isotope Evidence for the Utilization of Soil DIC by Karst Plants

Two issues needed to be addressed in field trials: (1) whether the natural abundance of $^{13}$C signals of soil DIC could be detected in leaves, and (2) what the major carbon source supplying the plant roots was. A few studies showed that even if the plants were irrigated with $^{13}$CO$_2$ or H$^{13}$CO$_3^-$ solution, the C isotope signals were too weak to be detected [20,57]. This is probably due to a low concentration of the labeled species dissolved in the solution (usually less than 1 mM), resulting in less $^{13}$CO$_2$ or H$^{13}$CO$_3^-$

taken up by roots as the diffusion against the concentration gradient from roots to the soil was minimal [15]. However, in the karst habitats, the high concentration of soil DIC made it possible to be fixed by plants. In addition, labeling experiments usually have a single type of carbon source [16,17,20], whereas, in field conditions, the major form of carbon source depends on soil pH, humidity, temperature, root respiration, microbial activity, etc. [58,59]. For this reason, we measured both the concentrations of soil DIC and $CO_2$ (Figure 2). The result suggested that soil DIC rather than soil $CO_2$ was the main source supplying the roots. However, direct evidence is needed to support this idea, especially in field conditions. Then we examined whether soil DIC could affect the $\delta_{WSOM}$. The experimental data confirmed our speculation that $\delta_{WSOM}$ largely deviated from the predicted $\delta^{13}C$ of photosynthates ($\delta_A$) solely assimilated from atmospheric $CO_2$ across nine species (Figure 3), consistent with our previous finding [26]. As the values of $\delta_A$-$\delta_{WSOM}$ were all positive and half of them were larger than 2‰ (Figures 3B and 5), the isotope measurement error (less than 0.1‰) and diurnal variations in $\delta_{WSOM}$ (less than 1‰) [60] are not sufficient to explain the discrepancy. When combined with the very negative $^{13}C$ signal of soil DIC ($\delta_A{}'$, see Supplementary Data), this large and positive deviation of $\delta_A$ from $\delta_{WSOM}$ suggests the participation of soil DIC in leaf photosynthesis. Therefore, both atmospheric $CO_2$ and soil DIC contributed to the newly assimilated carbon (i.e., leaf WSOM). The contribution of soil DIC ($f_{DIC\_soil}$) to leaf total photosynthesis could be quantified with a two-end-member mixing model (Equation (13)). We further showed that $\delta_A$-$\delta_{WSOM}$ was closely linked to $f_{DIC\_soil}$ among all species (Figure 5). That is, the more the soil DIC was involved in photosynthesis, the more $\delta_A$ deviated from $\delta_{WSOM}$, and the larger the proportion of soil DIC contributed to leaf WSOM.

However, uncertainties remain concerning the estimation of $f_{DIC\_soil}$. For example, we considered the tissues-respired $CO_2$ dissolved in the xylem sap in the calculation of $f_{DIC\_soil}$. Although we assigned a mean value of 5 mM for the $C_{\_R}$, species might differ in $C_{\_R}$ due to their different physiological statues and anatomical characteristics [15,52,61]. We also assumed that the source of respired $CO_2$ came from the newly assimilated carbon and used the value of $\delta_{WSOM}$ to infer the $\delta^{13}C$ of $CO_2$ respired by living tissues ($\delta_R$). Nevertheless, recent studies [62–65] showed that vegetation pools respired carbon of a wide range of ages in many species. For instance, the respired carbon in stems and roots was, on average, older than 1 year [65,66]. When plants were exposed to stress, respiration almost completely relied on the old carbon [64]. Therefore, in this study, there was a high probability that the investigated species used reserved carbon for respiration under the condition of frequent water limitation in karst habitats. To the best of our knowledge, the $\delta^{13}C$ of reserved carbon has a close relationship with that of respired $CO_2$ [56]. Our unpublished data showed that the maximum range of $\delta^{13}C$ in the storage carbon of some karst-adaptable species was 4.13‰ during the growing season, which implied a similar or less variation in the range of $\delta^{13}C$ for respired $CO_2$ in stems and roots. Some studies also showed that seasonal or annual variability of $\delta^{13}C$ in respired $CO_2$ in stems and roots ranged within 2.0–4.5‰ [67–69]. In this study, taking *L. lucidum* for example, a change of 4.5‰ in $\delta^{13}C$ of leaf WSOM only led to 0.5% of the change in $f_{DIC\_soil}$, indicating a limited influence of the source of carbon used for respiration on the quantification of plants' utilization of soil DIC.

### 4.2. Interspecies Difference in Plants' Use of Soil DIC

An assessment of $f_{DIC\_soil}$ between two plant life forms, namely trees and shrubs, showed no clear trend (Figure 4), indicating that the variation of $f_{DIC\_soil}$ was mainly resulted from the interspecies differences. In most species, the values of $f_{DIC\_soil}$ were comparable with those reported in the laboratory experiments [24–26], suggesting that soil DIC was easily accessible to plants in the karst habitats. In this study, nine species were all native, most of which were deciduous trees or shrubs. Although species differed in height, biomass, ages, etc., our measurements and sampling were all conducted with newly expanded leaves and within the root zones of the same soil layer. As leaves and roots control the uptake of atmospheric $CO_2$ and soil DIC, respectively, the species-specific

variation in $f_{\mathrm{DIC\_soil}}$ might be related to leaf gas-exchange traits (Table 2) and soil conditions (Figure 2).

$f_{\mathrm{DIC\_soil}}$ positively correlates with $A$ ($p = 0.003$) and $WUE_i$ ($p < 0.001$) but negatively with $c_i/c_a$ ($p < 0.001$). At first sight, it was surprising that $f_{\mathrm{DIC\_soil}}$ increased with an increase in $A$, which was inconsistent with the results reported by our previous study [26]. For instance, when plants were confronted with moderate or severe water limitation, it would drastically reduce $A$ and $g_s$ [7,29] and thus increased the proportion of root or soil-derived DIC to support photosynthesis [26,29]. The fact was that the change of $A$ was not proportional to that of $A_{\mathrm{DIC\_soil}}$ or $E$ [26], thus arithmetically promoting the value of $f_{\mathrm{DIC\_soil}}$. In this study, the level of $g_s$ indicated mild drought stress in many species (Table 2). Among these species, higher $A$ corresponded to higher $g_s$ and $E$, which benefited the long-distance transport of DIC in the xylem sap [70,71]. Similarly, higher $WUE_i$ also implied a higher contribution of soil DIC to leaf photosynthesis, although the correlation between $E$ and $f_{\mathrm{DIC\_soil}}$ was not significant (Table 3). The negative correlation between $f_{\mathrm{DIC\_soil}}$ and $c_i/c_a$ corresponded to the reverse change of $f_{\mathrm{DIC\_soil}}$ and the contribution of atmospheric $CO_2$ to leaf total photosynthesis ($1$-$f_{\mathrm{DIC\_soil}}$) because higher $c_i/c_a$ usually suggested higher $\Delta^{13}C_{\mathrm{com}}$ (Equation (2)) and lower $A$ and $E$ [32]. In addition, there was no significant correlation between $f_{\mathrm{DIC\_soil}}$, $C_{\mathrm{\_DIC}}$, and $\delta_{\mathrm{DIC}}$. This could be explained by less variation of these indicators in comparison with that of leaf gas-exchange parameters among species (Figure 2; Table 2).

*4.3. Effect of Altitude on Plants' Use of DIC*

As the altitude increases, the number of tree species reduces and that of shrub species increases (Table 1), suggesting a change in the plant community. In this study, only three species presented in all plots and thus were available for evaluating the effect of altitude on $f_{\mathrm{DIC\_soil}}$ (Figure 6). Among these species, $f_{\mathrm{DIC\_soil}}$ of *L. lucidum* and *A. delavayana* increased from the lower to higher altitude, whereas in *V. dilatatum*, $f_{\mathrm{DIC\_soil}}$ decreased along the altitudes. The impact of altitude on $f_{\mathrm{DIC\_soil}}$ was a little bit confusing. However, altitude did not directly affect $f_{\mathrm{DIC\_soil}}$, but through imposing influences on site-characteristic related microclimate and soil conditions [35,72]. For example, the altitude could constrain daily mean temperature and vapor pressure deficit [72], and produce considerable variations in soil moisture and nutrient availability [35]. However, in the present study, the air temperature and relative humidity recorded by LI-6400 did not linearly change with altitude (see Supplementary Data) but exhibited a high degree of spatial heterogeneity. The reasons might be that the altitude gradient was not large enough, and some occasional factors, e.g., wind, shading, and vegetation coverage, could redistribute these resources, e.g., light and vapor [72,73]. Our previous study reported decreasing patterns for soil water content, organic matter, and some nutrients with the altitudes in the same area and same season of 2015 [34]. However, the increasing trend of $f_{\mathrm{DIC\_soil}}$ along the altitude gradient in *L. lucidum* and *A. delavayana* was opposite to that in *V. dilatatum* (Figure 6), implying that the differences in soil conditions could not solely explain the variation of $f_{\mathrm{DIC\_soil}}$. Therefore, we speculated that all these variabilities were combined to influence the leaf gas-exchange and subsequent estimation of $f_{\mathrm{DIC\_soil}}$. Future studies may choose a large range of environmental gradients and exclude the influences of some occasional factors.

**5. Conclusions**

In the present study, we aimed to investigate whether soil DIC could be assimilated through leaf photosynthesis in the karst habitats, and whether this process, if taken place, varied between species and altitudes. Firstly, we showed that all plots in the study sites had a high content of soil DIC in the root zones of nine species. This implied a high possibility that soil DIC could be fixed by karst plants through photosynthesis. Secondly, we observed that there were large discrepancies between the measured and predicted $\delta^{13}C$ of newly formed photosynthates ($\delta_A$-$\delta_{\mathrm{WSOM}}$), and this systematic difference could not be explained by measurement errors or diurnal variations in $\delta_{\mathrm{WSOM}}$ alone. Therefore, we accounted

for the involvement of soil DIC in photosynthesis and thus deviating $\delta_{WSOM}$ from $\delta_A$. Thirdly, we applied a two-end-member mixing model to estimate the contribution of soil DIC to leaf total photosynthesis ($f_{DIC\_soil}$). The values of $f_{DIC\_soil}$ largely differed among species and some of which were influenced by altitudes. The present study improved our understanding of how plants adapted to alkaline soil conditions of karst habitats and acquired additional carbon for growth.

**Supplementary Materials:** The following supporting information can be downloaded at: https://www.mdpi.com/article/10.3390/agronomy12102489/s1.

**Author Contributions:** Conceptualization, S.R. and Y.W.; experimentation, S.R.; writing, S.R.; data analysis, S.R. and Y.W.; review and editing, Y.W. All authors have read and agreed to the published version of the manuscript.

**Funding:** This study was supported by Support Plan Projects of Science and Technology of Guizhou Province (No. (2021)YB453) and the National Key Research and Development Program of China (No. 2021YFD1100300).

**Institutional Review Board Statement:** Not applicable.

**Informed Consent Statement:** Not applicable.

**Data Availability Statement:** All data related to this article can be found in the online version.

**Acknowledgments:** We deeply thank W. Lin, two anonymous reviewers for their valuable comments and suggestions that greatly improved the manuscript.

**Conflicts of Interest:** The authors declare no conflict of interest.

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
