# Peer review of "Species-Specific and Altitude-Induced Variation in Karst Plants’ Use of Soil Dissolved Inorganic Carbon"

_agronomy, doi:10.3390/agronomy12102489_

Round 1
Reviewer 1 Report
In their study, the authors aim to show and quantify the amount of soil dissolved inorganic carbon contributing to carbon fixation and assimilation in various tree and shrub species in an alkaline karst ecosystem. Using an involved empiric and modelling approach they find that soil derived inorganic carbon accounts for 2.5-10% of C fixed during photosynthesis with differences between species and altitudes that don’t follow a clear trend.
The authors raise a timely and interesting question of alternative C sources for woody species, especially under environmental stressors (drought or site-specific). While the introduction is nicely leading into the topic, the following chapters lack some clarity and – in my opinion – miss some important points, especially the discussion section. Most importantly, I think different potential C sources must be discussed in the paper. It has been shown that the age of respired C can vary by quite a bit (see comment by Sierra et al. 2022) and so I wonder if some of the C that is extracted using a water leaching method (as done here in this study) might be older and therefore comprise of different d13C values than the atmospheric d13C recorded on the same day as the leaves were sampled. Secondly, the reasons for differences between species and altitudes remain very vague and therefore appear almost random. More information can be given, e.g. are all the species native in this area, are they all deciduous, etc. Similarly, for the different altitudes, is there even a difference in conditions (temperature, moisture, soil properties, etc.). If there is no site-characteristic difference, why would we expect different trends in C uptake? I am not entirely sure about the interpretation of Figure 5. fDIC is calculated directly from the two parameters dA and dWSOM, so in a way this graph is showing the results of the two-end member mixing model, i.e. the more one end member (here dA) is deviating from the mixing (dWSOM), the more important the other end member (here d’A) gets. I would appreciate if the authors would elaborate a little on this figure.
In addition, I have several minor notes that should be taken into consideration:
· Slope: Although this term is used by some to describe differences in altitude, I think a more appropriate term for what the authors describe would be “altitude” or “elevation”.
· L. 53-57: Long sentence, could be rephrased for better clarity.
· L. 98-111: Description of the study sites should include differences between the altitudes. The authors argue a lot with drought and state (L. 107) that the sites have low soil water content. Although already published elsewhere, I think a little more background on the drought events (maybe a dry season?) would be informative for the paper.
· L. 124-136: What was the CO2 concentration?
· L. 149: I can imagine that soil conditions vary between the sites. Did you distinguish between different horizons, was the organic layer removed or only the litter? Please be more specific here.
· L. 157: Was the leaf material grinded before dissolving in water? How many leaves were measured per tree/shrub (also for the gas exchange measurements)?
· L. 173: Please state type of pH probe.
· Eq. 2: Can’t see all the symbols in the formula. Also some of the other formulas further down seem a little bit shifted.
· L. 214/215: You can’t state that this is rhizosphere soil. Rhizosphere is considered only the soil that attaches directly to the roots of a plant.
· L. 319: “Less than 1.5‰” - where is that shown? Not referenced.
· L. 333: Table 2 is mentioned here first. This should be moved to the results section.
· L. 334: What is the dominant species?
· L. 345: I don’t quite understand: If the plant is stressed, shouldn’t the xylem flow be slower if the stomata are closing? How would (more) DIC be transported to the leaves then?
· L. 366-369: How is it possible that C assimilation is higher on higher altitudes for some plants and lower for others? If you argue with higher light availability, shouldn’t all plants have a similar trend?
· Table 2: Please add units to the table.
· Table 3: This table is never referenced in the manuscript. Should be mentioned and described in the results section.
Sierra C.A., Ceballos-Núñez V., Hartmann H., Herrera-Ramírez D. & Metzler H. (2022) Ideas and perspectives: Allocation of carbon from Net Primary Production in models is inconsistent with observations of the age of respired carbon. Biogeosciences 19, 3727–3738.
Reviewer 2 Report
Please follow the below-mentioned comments to improve the quality of the paper:-
Line No. 24, Remove the word “Karst” as it is mentioned in the Title.
Introduction
Line 50-51. The additional carbon gain may compensate for the decrease in the fixation of atmospheric CO2 induced by drought stress; add a latest reference to strengthen your thought.
Line 53-57: Despite extensive hydroponic experiments designed to simulate karst soil conditions, for example, bicarbonate stress (usually more than 10 mM), osmotic stress (simulating water scarcity), and their interactions [22‒24], it does not necessarily reflect the utilization of soil DIC by plants due to the differences in rhizosphere microenvironment. Hence, there is a need to fill this knowledge gap between laboratory experiments and field trials. I will suggest separating the references after every example, e.g., bicarbonate stress. osmotic stress? And finally, end this sentence with the latest reference to support your theme.
Line 59 – 65: Understanding the plants’ use of soil DIC relies on appropriate methods and models for quantification. Currently, there are two ways to estimate this utilization: (i) applying
the high abundance 13C labeling and then calculating the ratio of soil DIC fixed in specific
tissues or organs to the total carbon gain, which latter includes soil DIC fixation, ap-
parent photosynthesis measured by commercial Infrared Gas Analyzers, and respiration
[20], however, this calculation is not directly linked to photosynthesis; (ii) using two-
source H13CO3− labeling in combination with some isotope mixing models to determine
the contribution of root-derived DIC to leaf total photosynthesis [22‒24]. My suggestion is to break this sentence to make it meaningful; otherwise, it is vague and long-winded. And support your thoughts with the latest references.
Line 95 to 96- Pl adds this line to the abstract and categorically write your objectives in these lines.
Line 108- 109- The thickness of the soil layer varied between 27 cm and 58 cm. It should be 27 and 58 cm
Line No 65. Please correct H13CO3− it.
Line No 81. Replace “[24,28] with [24-28],
Line No. 159. Replace 12 000 g. with 12000 g.
Line No. 265. Replace “2‰” with 2%.
Line No. 309, 310, and 330. [16,17,20], [22–24]. Please adopt the relevant style
Line No. 334 please recheck it.( p < 0.01).
References
Select an appropriate or unique style for citation according to the journal policy and use the latest references, please.
My suggestion is major revisions.
Round 2
Reviewer 2 Report
Dear Authors
I am happy to see revised version. All changes are incorporated. Yet English language needs attention.
Regards
Author Response
Thank you very much. The English language and style have been edited in minor revised MS.